# OsMADS58 Stabilizes Gene Regulatory Circuits during Rice Stamen Development

**DOI:** 10.3390/plants11212899

**Published:** 2022-10-28

**Authors:** Liping Shen, Feng Tian, Zhukuan Cheng, Qiang Zhao, Qi Feng, Yan Zhao, Bin Han, Yuhan Fang, Yanan Lin, Rui Chen, Donghui Wang, Wenfeng Sun, Jiaqi Sun, Hongyun Zeng, Nan Yao, Ge Gao, Jingchu Luo, Zhihong Xu, Shunong Bai

**Affiliations:** 1State Key Laboratory of Protein & Plant Gene Research, Peking University, Beijing 100871, China; 2Key Laboratory of Plant Molecular Physiology, Institute of Botany, Chinese Academy of Sciences, Beijing 100093, China; 3College of Life Sciences, Peking University, Beijing 100871, China; 4Biomedical Pioneering Innovation Center (BIOPIC), Beijing Advanced Innovation Center for Genomics (ICG), Center for Bioinformatics (CBI), Peking University, Beijing 100871, China; 5Center for Bioinformatics, Peking University, Beijing 100871, China; 6Jiangsu Co-Innovation Center for Modern Production Technology of Grain Crops, Yangzhou University, Yangzhou 225009, China; 7National Center for Gene Research, State Key Laboratory of Plant Molecular Genetics, CAS Center of Excellence in Molecular Plant Sciences, Institute of Plant Physiology and Ecology, Chinese Academy of Sciences, Shanghai 200233, China; 8Novogene, Beijing 100015, China; 9School of Life Science, Sun Yat-sen University, Guangzhou 510260, China; 10Center of Quantitative Biology, Peking University, Beijing 100871, China

**Keywords:** OsMADS58, rice stamen, meiosis, tapetum, gene regulatory circuits

## Abstract

Rice (*Oryza sativa*) OsMADS58 is a C-class MADS box protein, and characterization of a transposon insertion mutant *osmads58* suggested that OsMADS58 plays a role in stamen development. However, as no null mutation has been obtained, its role has remained unclear. Here, we report that the CRISPR knockout mutant *osmads58* exhibits complex altered phenotypes, including anomalous diploid germ cells, aberrant meiosis, and delayed tapetum degeneration. This CRISPR mutant line exhibited stronger changes in expression of OsMADS58 target genes compared with the *osmads58* dSpm (transposon insertion) line, along with changes in multiple pathways related to early stamen development. Notably, transcriptional regulatory circuits in young panicles covering the stamen at stages 4–6 were substantially altered in the CRISPR line compared to the dSpm line. These findings strongly suggest that the pleiotropic effects of OsMADS58 on stamen development derive from a potential role in stabilizing gene regulatory circuits during early stamen development. Thus, this work opens new avenues for viewing and deciphering the regulatory mechanisms of early stamen development from a network perspective.

## 1. Introduction

The stamen plays a key role in the angiosperm life cycle by producing meiotic cells, nursing the developing pollen, and facilitating pollen dissemination. To fulfil these functions, the stamen primordia must diverge from cells differentiating into photosynthetic tissues, somatic cells in the primordia must differentiate into diploid germ cells that will commit to a meiotic fate through differentiation of microspore mother cells (MMCs), and specialized tissues such as the tapetum must differentiate to nurse pollen generation.

The ABC model for determining floral organ identity was first proposed using Arabidopsis and Antirrhinum as model plants and later shown to be generally applicable to other angiosperms including rice [1,2]. According to the ABC model, stamen identity is determined by the interaction of B-class genes, such as *APETALA3* (*AP3*)/*PISTILATA* (*PI*), and C-class genes, such as *AGAMOUS* (*AG*) in Arabidopsis [3,4,5,6]. 

In rice, the two MADS box genes *OsMADS3* and *OsMADS58* were identified as C-class genes [7]. Differing from the clear phenotype generated by loss of *AG* in Arabidopsis, although both genes are expressed in early stamen primordia [7,8], mutation of *OsMADS3* causes complicated alterations of organ types and numbers; stamens are still produced [7,8,9], but with abnormal development at later stages [10]. By contrast, mutation of *OsMADS58* (generated by transposon insertion) alone causes no obvious change in phenotype [8]. Double mutant plants for the two genes exhibit loss of stamens and carpels [8,11], similar to the phenotype of *ag* mutants [3]. A gene expression profiling study demonstrated that OsMADS58 functions as a transcription factor to bind target genes, affect their expression, and result in alteration of cellular redox status; notably, it was also revealed that the mutant previously used was not a null mutant [12]. Although Sugiyama et al. [11] generated TILLING and CRISPR (editing at exon 3) mutants, they observed no altered phenotypes. The lack of null mutations for *OsMADS58* have thus hampered efforts to fully elucidate its function.

The identification of *SPOROCYTELESS* (*SPL*) in Arabidopsis opened an avenue to decipher the mechanism of the transition from somatic cells to MMCs [13,14]. Intriguingly, Ito et al. [15] reported that transcription of *SPL* was directly regulated by the C-class MADS box protein AG, a finding that linked stamen organ identity determination and germ cell induction. Recently, Ren et al. [16] identified *OsSPL*, the rice homolog of Arabidopsis *SPL*, not only in sequence, but also in function. However, as no null mutant of *OsMADS58* was available, there was no clear way to investigate whether a similar regulatory mechanism of AG-to-SPL exists in rice.

Tapetum differentiation is regulated by many genes in rice [17,18]. Notably, mutants with aberrant MMC differentiation also fail to differentiate the tapetum properly, similar to that found in Arabidopsis [19,20,21,22]. It has therefore been suggested that tapetum differentiation is dependent on MMC differentiation [17,23].

To clarify the function of OsMADS58, we used CRISPR to generate a knockout mutant of *OsMADS58* and identified differences in phenotype and the transcriptional regulatory network between the knockout and transposon insertion alleles. These findings suggest that OsMADS58 is required for stabilizing gene regulatory circuits underlying the subsequent stamen development in rice.

## 2. Results

### 2.1. Osmads58 CRISPR Lines Exhibit Sterility

To create a null mutation for clarifying the function of *OsMADS58*, we generated CRISPR lines using the CRISPR-Cas9 system [24], targeting at exon 2. Among the different transgenic lines, three independent editions were found, two with a single base insertion (A at 56 bp and C at 56 bp, respectively) and one with a 4-bp deletion, all resulting in a frameshift (Appendix A). To assess any off-target changes, the genome of the CRISPR line with A insertion at 56 bp (designated *osmads58* CRISPR, the line first obtained with a sterile phenotype) was re-sequenced. Genomic sequence and RNA-seq data confirmed that the *OsMADS58* locus was edited as intended. While one SNP (A-G) and two indel candidates were identified in the genomic sequence (Appendix A), the corresponding three genes were either not expressed or showed no difference in expression in the tissues we are studying. To further explore whether the CRISPR line was indeed a null mutation of *OsMADS58*, we performed mass spectrum analysis of in vitro-translated OsMADS58 protein. No hallmark OsMADS58 peptides were identified (Appendix A). These data demonstrated that *osmads58* CRISPR is a null mutant of the *OsMADS58* gene.

While all three independent editions of CRISPR lines exhibit sterility in their homozygous status, we mainly used *osmads58* CRISPR for detailed analysis. No morphological aberrations were observed in the CRISPR lines during vegetative growth. However, stamen alterations were observed at the flowering stage. Unlike the previously described *osmads58* (renamed here as *osmads58* dSpm to distinguish it from *osmads58* CRISPR) lines, which remains fertile (Appendix A), the *osmads58* CRISPR plants were sterile, with no I_2_-IK staining observed in pollen and no seed-set at maturity (Figure 1). These data highlight that stamen development in the CRISPR lines was severely affected.

### 2.2. Both MMCs and Tapetum Are Aberrant in the osmads58 CRISPR Line

To clarify how the stamen development is affected in the *osmads58* CRISPR lines, we analyzed semi-thin sections of stamens throughout development. Before stage 4 [12,25], there was no distinguishable difference in cellular differentiation between the CRISPR lines and the wild type (Figure 2). After stage 5, morphological differences were observed, including increased numbers of MMCs from an average of 4.2 in the wild type to 5.5–6.2 in different independent CRISPR lines (Stage 5, *p* < 0.01. Figure 3A–E), delayed tapetum degeneration (Figure 2N), and abnormal microspore differentiation (Figure 2N–P). In brief, both MMCs and tapetum were aberrant.

#### 2.2.1. Disturbed Chromosome Behavior in Meiosis

The MMCs are the cells destined to undergo meiosis. To identify the consequences of the increase in MMC numbers (Figure 3A–E), we observed the meiosis behavior of *osmads58* CRISPR. We determined that the MMCs could enter meiosis, but many univalents could be detected (Figure 3G).

#### 2.2.2. Delayed Tapetum Degeneration

To further explore the delay of tapetum degeneration observed in semi-thin sections (Figure 2N), we performed TUNEL assays, which detect DNA damage in cells and are often used to monitor tapetum degeneration [26]. We observed that while the tapetum degeneration, indicated by the green-dotted signals, showed no distinguishable difference between the *osmads58* dSpm line and the wild type, it was significantly delayed in the *osmads58* CRISPR lines (Figure 4). Together with the histological observations from the semi-thin sections, these results demonstrate that tapetum degradation is delayed in the absence of functional OsMADS58.

### 2.3. Gene Expression Is Severely Disturbed in the osmads58 CRISPR Line

Previously, we reported that OsMADS58 protein can bind ~610 genes, including nucleus-encoded photosynthesis-related genes [12]. However, no genes with known functions in either the MMC differentiation (including meiosis) or tapetum differentiation were among the known OsMADS58 targets. We therefore aimed to address how the null mutation of this C-class MADS box gene caused aberrant MMCs and tapetum cells. We first performed qPCR to verify whether the typical OsMADS58 binding targets changed their expression levels in the *osmads58* CRISPR line. Then, we conducted RNA-seq analysis to observe the change in gene expression profile in the *osmads58* CRISPR and *osmads58* dSpm lines.

#### 2.3.1. OsMADS58 Target Genes Have Higher Expression in the *osmads58* CRISPR Line

Using the genes upregulated in the *osmads58* dSpm samples (Figure 7 in [12]) as targets, we examined OsMADS58 target gene expression levels in the *osmads58* CRISPR samples. The expression levels of these target genes in the *osmads58* CRISPR samples were not only higher than those in the wild type, but higher than those in the *osmads58* dSpm samples (Figure 5). This result is not only consistent with the previous inference that OsMADS58 inhibits the expression of its binding targets [12], but also suggests that such an inhibition is dependent on the quantity of OsMADS58 protein.

#### 2.3.2. Null Mutation of OsMADS58 Causes Vast Variation in Gene Expression Profiles

To explore the underlying molecular link between the loss of OsMADS58 function and the mutant phenotypes, we performed RNA-seq analysis with young panicles (about 1 cm in length, in which the stamens develop to stages 4–6, [12,25]) collected from the wild type, dSpm lines, and CRISPR lines. The gene expression data generated from the nine samples (including three, four, and two samples for wild-type, *osmads58* dSpm, and *osmads58* CRISPR lines, respectively) totaled over 420 million reads.

We identified more up- and downregulated genes in *osmads58* CRISPR than in the *osmads58* dSpm compared to the wild-type samples (Figure 6A,B). Thus, the knockout of OsMADS58 in *osmads58* CRISPR leads to more extensive alteration of downstream gene expression than that in *osmads58* dSpm, in which the OsMADS58 expression is simply downregulated. In addition, more than 85% of the differentially expressed genes (especially for downregulated genes) were specifically changed in either *osmads58* dSpm or *osmads58* CRISPR samples. Moreover, 74 and 96 of the 610 target genes of OsMADS58 in ChIP-seq data [12] were significantly differential expression in *osmads58* dSpm and *osmads58* CRISPR, respectively.

GO analyses revealed that in the *osmads58* dSpm samples, the genes upregulated compared to the wild type were mainly enriched for some basic metabolic pathways, such as microtubule-based movement. The genes downregulated were barely enriched in only two GO pathways (Figure 6C,D): response to stimulus and regulation of biological quality. By contrast, in the *osmads58* CRISPR samples, the genes upregulated compared to the wild type were enriched for GO pathways involved in cellular behavior such as microtubule and cell division. The genes downregulated were preferentially enriched for stimulus responses, especially to oxygen-containing compounds (Figure 6E,F).

#### 2.3.3. Null Mutation of OsMADS58 Markedly Alters the Expression of Genes in Key Functional Pathways in Stamen Development

We further analyzed the genes cataloged in functional pathways necessary for stamen development (Figure 6E,F). Gene expression patterns were clearly distinguishable between the wild-type, *osmads58* dSpm, and *osmads58* CRISPR samples in the pathways of chloroplast development, oxidation reduction, cell cycle, meiosis, and tapetum differentiation (Appendix A). The *osmads58* dSpm samples overall patterns in the pathways of oxidation reduction, cell cycle, and meiosis were more similar to the wild type than to the *osmads58* CRISPR samples (Figure 7A). This difference might help to explain why the pleiotropic phenotypes of *osmads58* CRISPR lines are more severe than those of the *osmads58* dSpm lines.

To validate the altered expression revealed by RNA-seq, the expression levels of representative genes were examined by RT-qPCR. The changes in gene expression level observed in the RNA-seq analysis were consistent with those observed in the RT-qPCR analysis (Figure 7B).

### 2.4. Correlations between Altered Developmental Events and Gene Expression Networks

The above results demonstrated that the null mutation of OsMADS58 caused aberrant MMCs and tapetum cells. However, we did not identify genes with known functions in MMC and tapetum differentiation among the direct binding targets of OsMADS58 [12]. In addition, many genes showed changes in expression levels, both higher and lower, in the null mutants. We therefore sought to elucidate what role OsMADS58 could play in MMC and tapetum differentiation and via what mechanism.

Screening suppression mutants to recover genes activated downstream of target genes can reveal linear regulatory pathways that explain the mechanism of the original mutant gene. While such an approach works in many cases, the advent ChIP-chip and ChIP-seq technology revealed that individual transcription factors often bind to and regulate hundreds, or even thousands, of genes. Therefore, it can be difficult to assess the full role of a transcription factor through analysis of gene–gene interactions one by one. Here, we used gene expression data at the genomic level and network analysis to explore how the loss of OsMADS58 causes the aberrant MMCs and tapetum cells.

#### 2.4.1. Construction of a Transcriptional Regulatory Network (TRN) Centered on OsMADS58

To construct the gene regulatory network driven by OsMADS58, we focused on the differentially expressed genes between the wild type and OsMADS58 expression perturbation (*osmads58* dSpm and *osmads58* CRISPR, respectively). The regulation edges where both transcription factors and corresponding target genes are differentially expressed were extracted from the regulation data in PlantRegMap [27]. Meanwhile, the regulation edges between OsMADS58 and its direct target genes identified by ChIP-seq [12] were obtained. These two types of regulation edges were merged into putative regulatory interactions. Considering that the downstream genes of OsMADS58 are supposed to be regulated due to a change in OsMADS58 expression, we only retained the regulatory interactions where the TF and corresponding target genes showed a strong (positive or negative) expression correlation across all samples (Figure 8A,B). The result showed distinct patterns in gene regulation generated in these two mutant lines (*osmads58* dSpm and *osmads58* CRISPR). For example, for the direct target genes of OsMADS58, there were more upregulated genes than downregulated ones in the *osmads58* dSpm line, i.e., 110 genes were upregulated in the *osmads58* dSpm line over wild-type levels, while only 24 genes were downregulated. This pattern is consistent to the severity of phenotypes of the two mutant lines described above.

#### 2.4.2. Alteration of TRNs during Stamen Development

Since we determined that OsMADS58 can bind genes, including those encoding photosystem components, and affect chloroplast development [12], and that the *osmads58* CRISPR mutant resulted in aberrant MMC (cell cycle), meiosis, and tapetum differentiation, we particularly focused on TRNs correlated to these five biological events, i.e., chloroplast development, redox status, cell cycle, meiosis, and tapetum differentiation.

Following a similar pipeline for TRN construction as above, we used RNA-seq data (including all differentially expressed genes, not just the direct binding targets of OsMADS58) to construct sub-TRNs of the genes involved in the above-mentioned five developmental events (Figure 8C–L). In comparing the sub-TRNs of the *osmads58* dSpm and *osmads58* CRISPR lines, we observed that in agreement with the greater number of genes with altered expression in the *osmads58* CRISPR lines than in the *osmads58* dSpm lines, more regulated connections (links in the diagram) were predicted in the *osmads58* CRISPR lines. This pattern was also consistent with the TRN constructed with the differentially expressed direct-binding targets of OsMADS58 (Figure 8A,B).

Among the numerous differences in network structure (i.e., the number of genes in networks) between *osmads58* dSpm and *osmads58* CRISPR, one phenomenon was striking: Unlike the other four sub-TRNs, in which OsMADS58 (LOC_Os5g11414) was the central hub, the sub-TRN for chloroplast development had LOC_Os3g64260 (an AP2 domain gene) as the central hub (Figure 8D) in the *osmads58* CRISPR lines. By contrast, there was no differential expression of LOC_Os3g64260 in the *osmads58* dSpm lines compared to the wild type (Figure 8C). These observations reveal that the TRN analysis provided a link between one gene loss and its pleotropic morphological consequence.

#### 2.4.3. Alteration of the Regulatory Circuits Underlying the Meiosis Aberrant

Meiosis is a developmental event that has been well investigated, with relatively rich information. Univalent phenomenon observed in the *osmads58* CRISPR line has been reported in the rice mutants in the meiotic genes *OsSDS, CRC1, PAIR2, OsSPO11-1, OsMTOPVIB*, and *P31^comet^* [28,29,30,31,32,33]. As there were no off-target mutations in those genes (Appendix A) and none of these known meiotic genes is a direct target of OsMADS58, we aimed to learn what might cause the univalent phenomenon in the osmads58 CRISPR line. 

Based on the RNA-seq data, all of these known rice meiotic genes were expressed in the CRISPR line, but with altered levels (Figure 9A). RT-qPCR analysis verified that the expression of these genes was indeed significantly changed in the CRISPR line compared to the wild type (Figure 9B). Notably, *PAIR1* (LOC_Os03g01590), which is involved in meiosis, exhibited significant changes in expression level in the *osmads58* CRISPR line and was placed in the sub-TRN (red or blue dots and linked with lines, Figure 8J, indicated by the red circle) for meiosis from that line. By contrast, no significant changes in meiotic gene expression were observed in the *osmads58* dSpm line, and none of those genes were placed in the sub-TRN (gray dots and no lines linking them, Figure 8I). These results were consistent with the fact that the univalent formation was observed only in the *osmads58* CRISPR line. This finding suggests that while the univalent formation can be caused by loss of function of particular genes, it can also be caused by alteration of regulatory circuits consisting of a group of genes.

### 2.5. Network Topologies Explain the Phenotype Differences between dSpm and CRISPR Lines

Different methods of mutagenesis—and even the same method on the same gene—often generate phenotypes of differing severity, not to mention the confounding effects of pleiotropy. Such problems are difficult to solve with traditional genetic tools. However, genomic tools provide an opportunity to explore the underpinnings and mechanisms of such phenotypic variation.

Here, we took advantage of detailed characterizations of the two different lines of *OsMADS58* mutants, dSpm and CRISPR, both at the molecular and phenotypic levels. We noted that in addition to obvious differences between the two mutant lines in the TRNs of direct targets of OsMADS58 (Figure 8A,B), and the sub-TRNs for particular developmental events (Figure 8C–L), overall differences in the regulatory circuits associated with all five events were even more striking (Figure 10). The differences in the TRNs between the two lines are consistent with the severity of their mutant phenotypes, highlighting the effects of gene expression regulation circuits on the phenotype in rice.

## 3. Discussion

In this work, we successfully generated a null mutant for OsMADS58, which allowed us to determine that loss of this gene resulted in male sterility, an ultimate consequence of a series of aberrant events, including meiosis and tapetum differentiation.

To explore how the loss of function in the C-class MADS box protein OsMADS58 causes the complicated morphogenetic events, instead of looking for up- and down-stream interacting genes through screening more mutants, we analyzed the TRNs based on RNA-seq data. This approach revealed correlation between the regulatory circuits and some critical biological processes including meiosis. These results provided a relatively simple but reasonable explanation to the pleiotropic phenotype regarding early stamen development caused by the loss of function of OsMADS58. Based on these observations, we concluded that OsMADS58 may play a role in stabilizing the gene regulatory circuits in rice stamen development.

The relationship between a gene and its morphogenetic effects might be complicated. Therefore, traditionally, scientists often used morphogenetic effects as a trait to represent the gene function, which led to many great findings in the pre-genomic era. However, recent findings reveal that genes used considered as key regulator of a particular trait may not origin for the trait, but were coopted into the role during trait emergence and responsible to maintain the trait. One of the recent example is that *LEC1 (LEAFY COTYLEDON1)* gene was considered as a key regulator for seed development but then found its origin and function in pteridophytes [34,35]. A similar situation was reported in animals, such as genes known to be used for nerve cells found in sponge [36]. From this perspective, it is easy to understand why loss of function of OsMADS58 did not exhibit phenotype of abnormal floral organ identity even though according to sequence similarity, it belongs to the C-class genes in ABC model [7].

Along with the progress of new technologies and accumulation of gene sequencing and expression information, it become a new challenge on deciphering the relationship between the genes, which we can manipulate and the phenotypes, which are directly interested to researchers. Based on our findings of the TNR analysis, we found that network mediation between the gene and phenotype might be a plausible to interpret how a single gene knockout causes a pleiotropic phenotype with no direct regulation between the gene knocked out and genes with known functions in the phenotypes. Furthermore, such an approach identified more candidate genes could be examined for their potential roles in biological processes such as germ cell differentiation, meiosis, and tapetum degeneration, such as *OsAP2-like* (LOC_Os3g64260. Figure 8C,D), which is previous unknown of being involved in rice stamen development. These genes may not be identified of the relationship to the biological processes if only using conventional genetic approaches.

At this moment, the “network mediation” interpretation provides new perspectives to view biological complexity. In the short run, such an interpretation might be applied to analyzing other complicated biological processes, for instance, genes affecting developmental traits through redox status, for which there is increasing evidence [37,38,39]. However, as redox status is a result of complicated metabolic circuits, it is difficult to decipher the mechanism through conventional genetic approaches. The perspective of “network mediation” might provide an alternative approach. In the long run, the “network mediation” interpretation might explain functional differences among homologous genes in different species. A gene that is recruited to different circuits during evolution would cause a different downstream effect regardless of sequence similarity. Powered by the rapid development of sequence technology, as well as the tools of network analysis, “network mediation” might be a new horizon to explore.

## 4. Materials and Methods

### 4.1. Plant Materials and Growth Conditions

We used rice (*Oryza sativa*) plants including japonica rice cultivar Nipponbare and *osmads58* dSpm (with a dSpm insertion 117 bp after the 5′-splice site of the second intron), which were obtained as a gift from Dabing Zhang and Martin Kater, confirmed according to [8].

The *osmads58* CRISPR lines were generated using the CRISPR/Cas9 system. A 23-bp target sequence with the protospacer adjacent motif (PAM) was chosen and ligated into the vector pH-Ubi-cas9-7, and sequenced plasmids were transformed into rice through Agrobacterium-mediated transformation. The genomic region surrounding the CRISPR target site for *OsMADS58* was amplified by PCR and then sequenced to screen for mutations. The primers that were used are listed in Appendix A. About 30% of *osmads58* CRISPR homozygotes were completely sterile. The CRISPR materials used in this article are completely sterile plants.

The plants were grown in a paddy field during the normal growing season in Beijing, China, and in Hainan province in China during the winter. A growth chamber (30 °C 6 ± 2 °C, 11 h light/13 h dark) was also used for rice culture.

### 4.2. In Vitro Translation of OsMADS58 and Mass Spectrometry Analysis

Total RNA was isolated with a Plant RNA Kit (OMEGA, Norcross, GA, USA, #R6827) from the wild type, *osmads58* dSpm, and *osmads58* CRISPR lines. First-strand cDNA was synthesized with a qPCR RT Kit (TOYOBO, Osaka, Japan, #FSQ-101). The *OsMADS58* CDS sequences were amplified by PCR in the three samples. The *OsMADS58* RNA and protein in vitro synthesis assay was performed using the MEGAscript Kit (Life Technologies, Carlsbad, CA, USA, #AM1330) and Wheat Germ Extract Systems (Promega, Madison, WI, USA, #L4380), according to the manufacturer’s instructions.

After testing protein quality, the protein samples were digested with endoproteinase gluc. Peptides were separated by liquid chromatography and then analyzed on a Q Exactive HF-X mass spectrometer (Thermo Fisher, Waltham, MA, USA).

### 4.3. Genome Resequencing and SNP Calling of osmads58 CRISPR Lines

One rice wild-type (WT) and three *osmads58* CRISPR lines (L1, L2, and L3) were sequenced on the Illumina HiSeq2500 platform with the 2X150-bp paired-end module. About 50~80x of genome sequence was generated for each sample. Raw reads were screened by quality checking with FastQC https://www.bioinformatics.babraham.ac.uk/projects/fastqc/, accessed on 1 October 2022. All reads passing the quality check were then aligned against the O. sativa reference genome, IRGSP1.0 [40], using bowtie2-2.2.6 [41]. The alignments were sorted according to genomic coordinates with SAMtools4 [42], and duplicated reads were filtered with the Picard http://broadinstitute.github.io/picard/ package, accessed on 1 October 2022. Multi-sample SNP calling was performed using the Genome Analysis Toolkit (GATK, version 3.7.0) [43] and SAMtools. Only the SNPs detected by both methods were analyzed further. To remove unreliable SNPs, hard filtering was performed on the SNP calls, as suggested by GATK’s Best Practices.

### 4.4. Characterization of Mutant Phenotypes 

Whole plants and panicles following seed maturation of the wild type and *osmads58* were photographed with a Nikon digital camera. Florets were photographed using a ZEISS Lumar V12 stereomicroscope (Carl Zeiss, Oberkochen, Germany). Mature pollen grains were stained with I_2_-KI solution for a few minutes and then visualized using an Imager D2 microscope and a ZEISS AxioCam ICc5 digital camera (Carl Zeiss, Oberkochen, Germany).

For semi-thin sectioning, panicles were fixed in FAA fixative (50% ethanol, 10% formaldehyde, 5% acetic acid) overnight at 4 °C, and then the materials were dehydrated under a gradient series of ethanol (50%, 70%, 80%, 95%, and 100%). The panicles were transferred to 1:1 ethanol:LR white resin (*v*/*v*) overnight and then transferred to 100% LR white resin for 24 h. The samples polymerized for 24 h at 65 °C. The embedded samples were sectioned (4 μm thick) before staining with toluidine blue O (Urchem, Shanghai, China) and imaged using an Imager.D2 microscope with a ZEISS AxioCam ICc5 digital camera (Carl Zeiss, Oberkochen, Germany).

### 4.5. TUNEL Assay

Panicles were fixed in FAA fixative (50% ethanol, 10% formaldehyde, 5% acetic acid) overnight at 4 °C. After dehydration under a gradient series of ethanol and xylene, panicles were embedded in paraplast plus (Leica, Wetzlar, Germany) and sectioned to a thickness of 7 µm using the RM2245 rotary microtome. The paraffin sections were dewaxed in xylene and rehydrated in an ethanol series. The TUNEL assay was performed using the DeadEnd Fluorometric TUNEL system (Promega, Madison, MI, USA, #G3250), according to the manufacturer’s instructions. Signals were observed and imaged using an Imager D2 microscope (Carl Zeiss, Oberkochen, Germany).

### 4.6. Chromosome Spreads Staining

Young panicles of the wild type and *osmads58* CRISPR were fixed in Carney’s solution (ethanol:glacial acetic acid, 3:1). Microsporocytes undergoing meiosis were squashed on a clean slide and stained with acetocarmine solution. After washing with 45% acetic acid, the slides were frozen in liquid nitrogen and the cover slips were removed. Microsporocytes were counterstained with DAPI in an antifade solution.

### 4.7. Real-Time PCR Expression Assay

Total RNA was isolated with a Plant RNA Kit (OMEGA, Norcross, GA, USA, #R6827). First-strand cDNA was synthesized with a qPCR RT Kit (TOYOBO, Osaka, Japan, #FSQ-101) according to the manufacturer’s instructions. Real-time PCR analysis was performed using the ABI7500 machine and SYBR Premix Ex Taq Mix (Takara, Dalian, China, #RR420A). LOC_Os06g48970 was used as an internal reference as described previously [8]. The real-time PCR results were analyzed using ABI7500 analysis software. Each experiment had three biological replicates. Specific primers for each target gene are listed in Appendix A.

### 4.8. RNA Extraction, cDNA Library Preparation, and Sequencing

Total RNA was extracted from panicles using Trizol reagent (Invitrogen, Carlsbad, CA, USA) and purified using the RNeasy Plant Mini Kit (Qiagen, Hilden, Germany). RNA quality was checked with Qubit 3.0 and Agilent 2100 instruments. Sequencing libraries were prepared according to the manufacturer’s instructions (Illumina, San Diego, CA, USA). We used magnetic beads with Oligo (dT) to enrich the mRNA and then added fragmentation buffer to break the mRNA into short fragments. The fragments and random hexamer primers were used to synthesize first-strand cDNA, which was then transformed into double-stranded cDNA with RHase H and DNA polymerase I. The sequencing library was constructed via PCR amplification. AMPureXP beads were used to purify the PCR product to obtain a chain-specific cDNA library. The library was subjected to paired-end sequencing using the Illumina HiSeq platform.

### 4.9. Samples for RNA-Seq, Gene Expression Calculation and Differential Analysis

Samples for RNA-seq were collected from three wild type lines, four *osmads58* dSpm lines and two independent transgenic *osmads58* CRISPR lines, respectively. To ensure the samples collected from the *osmads58* CRISPR were homozygotic and male sterile, the young panicles were collected after the genotyping. Every plant collected 2 young panicles, and 8 panicles from 4 plants of each independent transgenic *osmads58* CRISPR lines were mixed as one sample for RNA seq. Therefore, for all three lines, i.e., wild type, *osmads58* dSpm, and *osmads58* CRISPR, each had 3, 4 and 2 biological replicates, respectively.

All nine samples generated 420 million reads in total. The reads from RNA-seq experiments were mapped into the rice genome (MSU v7.0) [40] using the guidance of gene annotation with Hisat (v2.1.0) [44] and the “--dta” option. The gene expression levels for each sample were calculated using Stringtie (v1.3.3b) [45] with the options of “-B -e”. The gene expression files for different samples were pasted into a gene expression matrix (gene * sample) for the subsequent analyses. Two sets of differentially expressed genes (WT vs. 58-dSpm and WT vs. 58-CRIS) were calculated using DESeq2 (v1.16.1) [46] with the cutoff of “adjusted *p*-value < 0.05 and abs (FoldChange) > 2”.

### 4.10. TRN Construction

The regulation data (from transcription factors to target genes) in rice were downloaded from PlantRegMap [27]. Then, the regulation data for which both transcription factors and corresponding target genes were differentially expressed were extracted to obtain a putative regulation set. The direct target genes of OsMADS58 were downloaded from a previous work [12], and the regulation relationship between OsMADS58 and differentially expressed direct target genes of OsMADS58 were added to the putative regulation set. For each regulation in the putative regulation set, the Spearman correlation between transcription factors and corresponding target was calculated, and only the regulation sets with correlation ≥ 0.25 or correlation ≤ −0.25 were retained.

## Figures and Tables

**Figure 1 plants-11-02899-f001:**
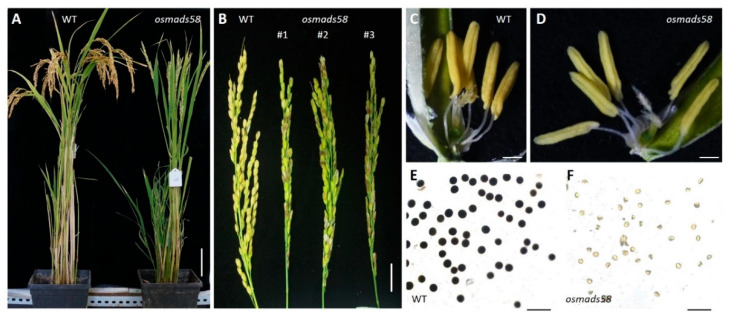
The *osmads58* CRISPR lines exhibit male sterility. (**A**) Comparison of wild-type (WT) and *osmads58* CRISPR plants after seed maturation. Bars, 10 cm. (**B**) Panicles in wild-type and *osmads58* CRISPR plants. Bars, 1 cm. (**C**,**D**) Stamens in wild-type (**C**) and *osmads58* CRISPR (**D**) plants; the wild type had normal yellow stamens, and the mutant appeared slightly yellow. Bars, 500 μm. (**E**,**F**) Wild-type (**E**) and *osmads58* CRISPR (**F**) pollen grains stained by I_2_-KI solution; *osmads58* CRISPR had severe defects in starch accumulation. Bars, 200 μm.

**Figure 2 plants-11-02899-f002:**
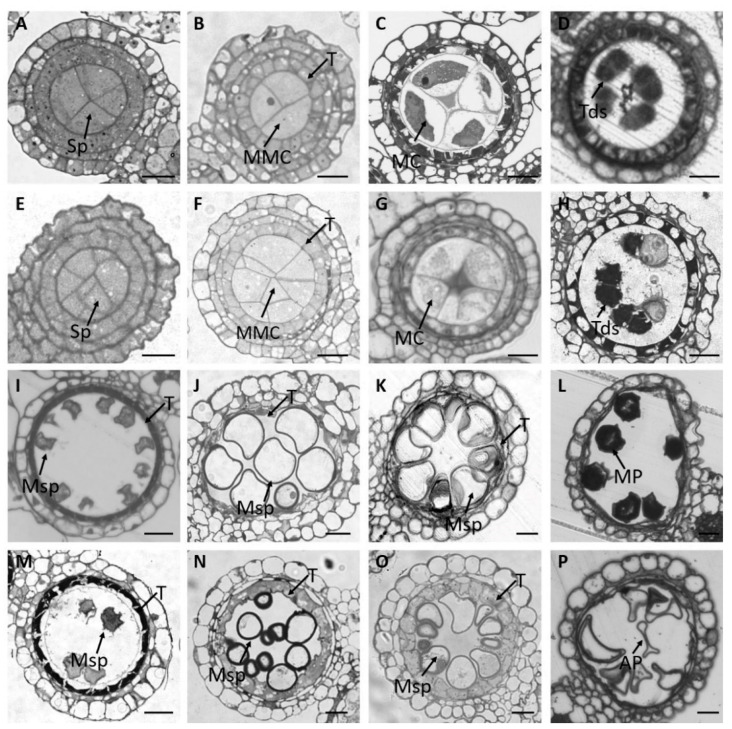
Abnormal development of microspore mother cells and the tapetum in *osmads58* CRISPR according to stamen semi-thin sections. (**A**–**P**) Semi-thin transverse sections of one lobe of anthers in the wild type (**A**–**D**,**I**–**L**) or *osmads58* CRISPR (**E**–**H**,**M**–**P**). (**A**,**E**) Anthers at stage 4. (**B**,**F**) Anthers at stage 5, there are more microspore mother cells in the mutant. (**C**,**G**) Anthers at stage 6. (**D**,**H**) Anthers at stage 8. (**I**,**M**) Anthers at stage 9. (**J**,**N**) Anthers at stage 10, the tapetum remains thick in the mutant line. (**K**,**O**) Anthers at stage 11. (**L**,**P**) Anthers at stage 12, abnormal pollen was observed in the mutant. Sp, sporogenous cells; MMC, microspore mother cell; MC, meiotic cell; Tds, tetrads; Msp, microspore; T, tapetum; MP, mature pollen; AP, abnormal pollen. Bars, 20 μm.

**Figure 3 plants-11-02899-f003:**
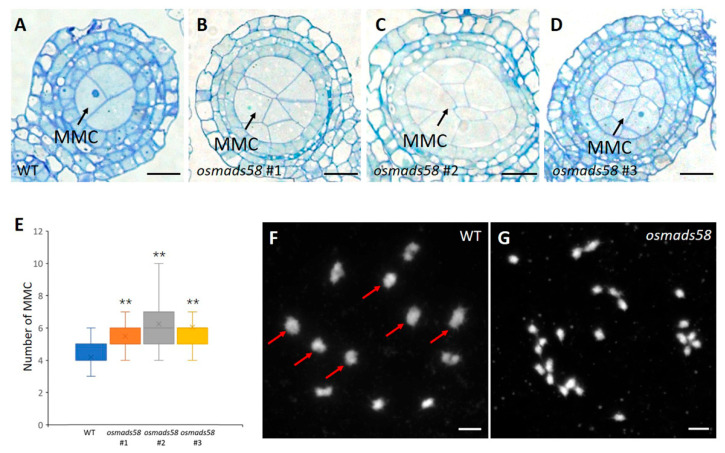
The number of microspore mother cells increases and the meiocytes exhibit severe defects in crossover formation in *osmads58* CRISPR lines. (**A**–**D**) Semi-thin transverse sections of one lobe of anthers in the wild type (**A**) and *osmads58* CRISPR (**B**–**D**) at stage 5. There were more microspore mother cells in all three lines of *osmads58* CRISPR compared to the wild type. Bars, 20 μm. MMC, microspore mother cell. (**E**) The number of microspore mother cells (MMC) in the wild type and three *osmads58* CRISPR lines. Error bars represent sd (*n* = 37). Student’s paired *t*-test: ** *p* < 0.01. (**F**,**G**) Comparison of chromosome behaviors in wild-type (**F**) and *osmads58* CRISPR (**G**) meiocytes at diakinesis, many univalents formed. The red arrows show normally paired chromosomes. Chromosomes were stained with 4′,6-diamidino-2-phenylindole (DAPI). Bars, 5 μm.

**Figure 4 plants-11-02899-f004:**
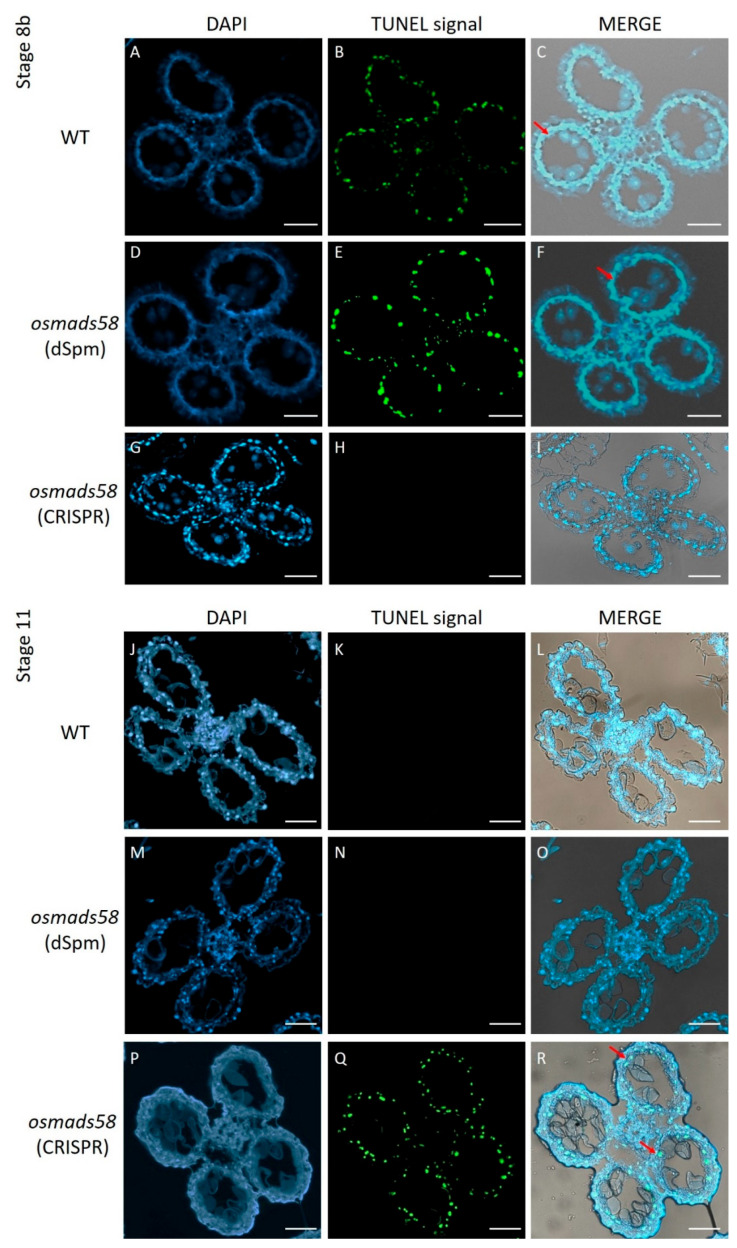
TUNEL assay of DNA fragmentation in tapetum tissue from wild-type, *osmads58* dSpm, and *osmads58* CRISPR lines. (**A**–**I**) Wild-type (**A**–**C**), *osmads58* dSpm (**D**–**F**), and *osmads58* CRISPR (**G**–**I**) anthers at stage 8. The tapetum of the wild type started to degrade, but *osmads58* CRISPR had no degradation signal at stage 8. (**J**–**R**) Wild-type (**J**–**L**), *osmads58* dSpm (**M**–**O**), and *osmads58* CRISPR (**P**–**R**) anthers at stage 11. The tapetum degradation of the wild type had completed, but the *osmads58* CRISPR still had degradation signals at stage 11. Blue signal indicates 4′,6-diamidino-2-phenylindole (DAPI) staining, while green fluorescence indicates a TUNEL positive signal. (**C**,**F**,**I**,**L**,**O**,**R**) Merged images of DAPI staining, TUNEL signal, and bright-field micrographs. The red arrows show the TUNEL positive signal in the tapetum. Bar, 50 µm.

**Figure 5 plants-11-02899-f005:**
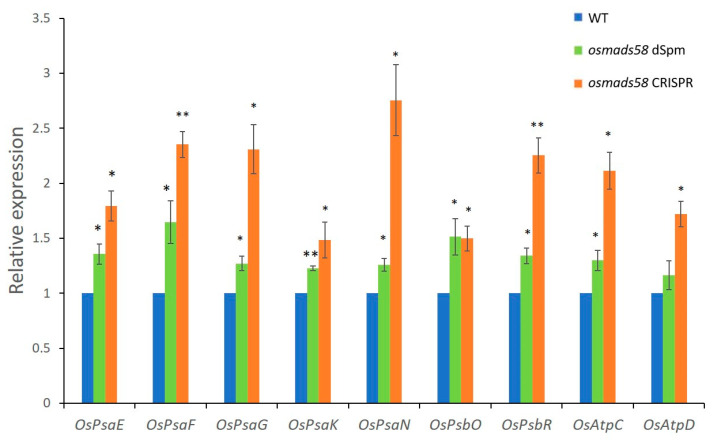
Expression levels verification of OsMADS58 target genes in *osmads58* dSpm and *osmads58* CRISPR lines. RT-qPCR analysis of expression levels of photosynthetic genes. These photosynthetic genes were upregulated in both *osmads58* dSpm and *osmads58* CRISPR lines, and the expression levels of these genes were much higher in *osmads58* CRISPR samples. Error bars indicate SD of three biological replicates. Student’s paired *t*-test: * *p* < 0.05, ** *p* < 0.01.

**Figure 6 plants-11-02899-f006:**
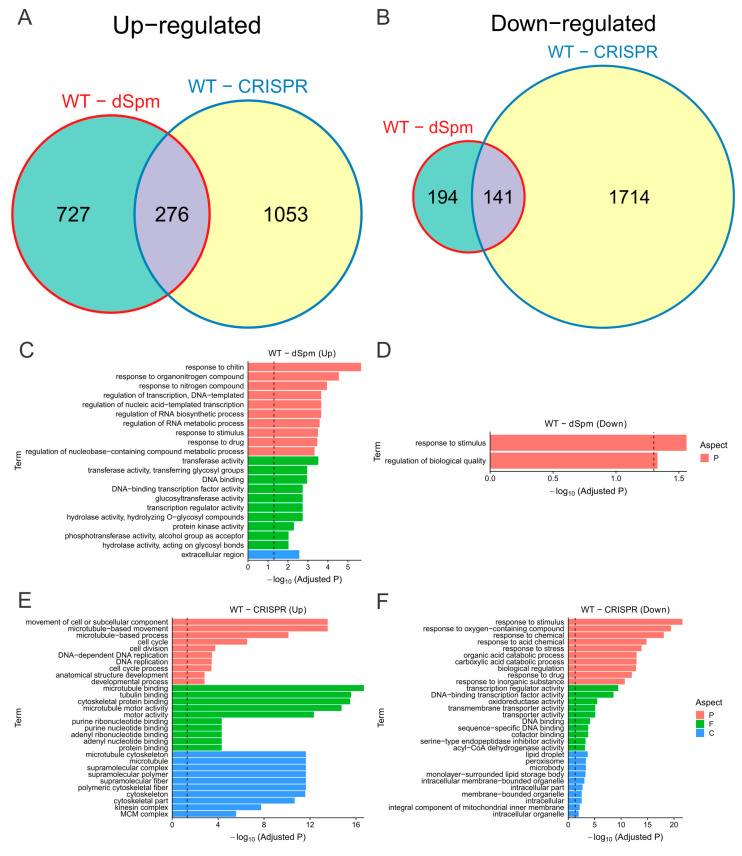
Comparison of gene expression changes during OsMADS58 expression perturbation. (**A**,**B**) Venn plots of differentially expressed genes (between wild type vs. *osmads58* dSpm and wild type vs. *osmads58* CRISPR). The upregulated (**A**) and downregulated (**B**) genes were used for comparison, respectively. The circle sizes denote the numbers of differentially expressed genes in the corresponding gene sets. (**C**–**F**) Enriched Gene Ontology (GO) terms for each set of differentially expressed genes. Colors denote the aspects of GO terms (P: biological process, F: molecular function, C: cellular component). Only the top 10 enriched terms are shown (Fisher’s exact test, *p* values adjusted by FDR).

**Figure 7 plants-11-02899-f007:**
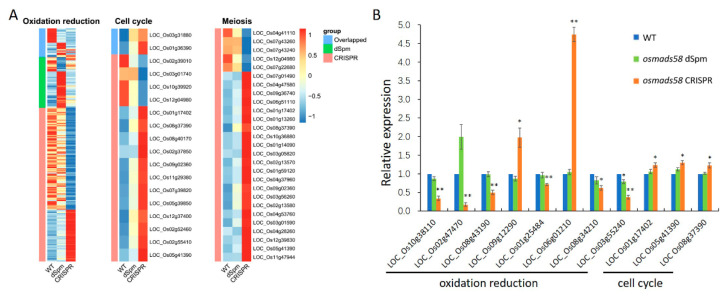
Expression patterns of genes in key pathways and selected genes expression levels verification by RT-qPCR. (**A**) Heatmaps of the expression levels of genes (rows) across conditions (columns). Only the differentially expressed genes are shown. Expression levels are averaged in each condition, followed by scaling in each row. The side color bars indicate whether the genes are differentially expressed in both two conditions or specifically differentially expressed in only one condition. (**B**) RT-qPCR analysis of expression levels of genes involved in oxidation reduction and cell cycle. Error bars indicate SD of three biological replicates. Student’s paired *t*-test: * *p* < 0.05, ** *p* < 0.01.

**Figure 8 plants-11-02899-f008:**
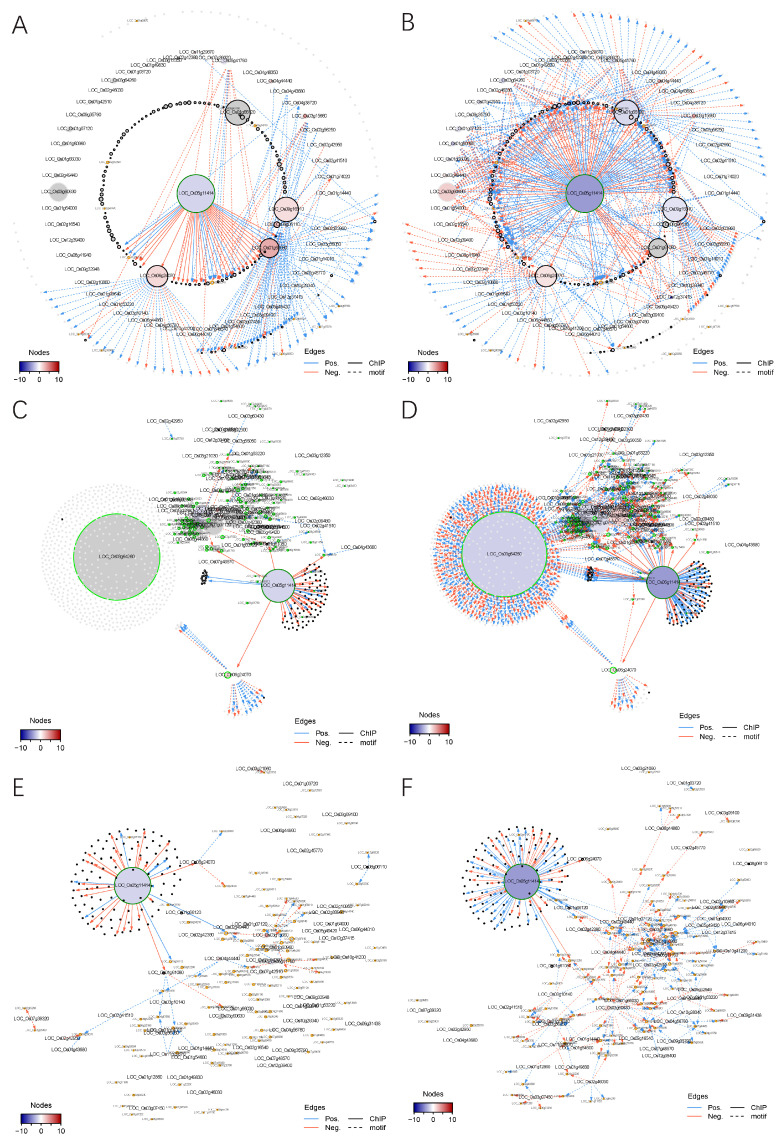
The sub-transcriptional regulatory network in different types of perturbation. (**A**,**B**) Transcriptional regulatory networks for the target genes of OsMADS58 in WT-dSpm (**A**) and WT-CRISPR (**B**) comparisons. Only the genes either differentially expressed in at least one comparison or directly targeted by OsMADS58 are shown. The node sizes indicate the number of neighborhoods for corresponding genes. *OsMADS58*, OsMADS58 target genes, and oxidation reduction-related genes are highlighted by green, black, and orange circles, respectively. Gene IDs of transcription factors and oxidation reduction-related genes are shown with big and small text size, respectively. Nodes are colored according to high (red) or low expression (blue) related to wild-type (WT) samples. Grey nodes indicate non-differentially expressed genes. Edges denote regulation supported by ChIP-seq data (solid) or predicted by motif scanning (dashed). Only the edges with an absolute of correlation greater than 0.25 are shown. Edges are colored according to positive (blue) or negative (red) correlation between gene pairs. (**C**–**L**) Transcriptional regulatory networks for each pathway in WT-dSpm (left) and WT-CRISPR (right) comparisons. (**C**,**D**) Chloroplast development genes; (**E**,**F**) oxidation reduction genes; (**G**,**H**) cell cycle genes; (**I**,**J**) meiosis genes; (**K**,**L**) tapetum differentiation genes. Only the upstream and downstream genes of the pathway-related genes differentially expressed in at least one comparison are shown. High quality original files (**A**–**L**) can be found in the Appendix A.

**Figure 9 plants-11-02899-f009:**
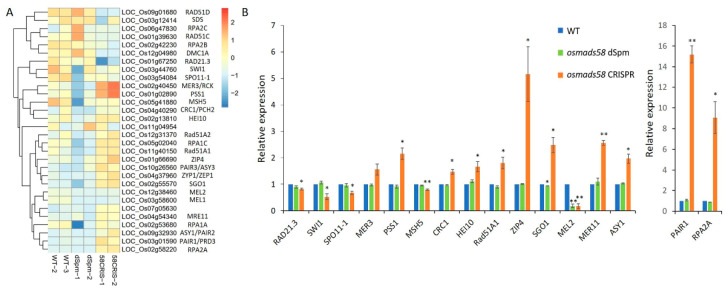
Expression profiles of meiotic genes in different samples and selected genes expression levels verification by RT-qPCR. (**A**) Known meiosis-related genes expression levels from the RNA-seq data. (**B**) RT-qPCR verification of the expression levels of selected meiotic genes. To better present the differences, the expression of two genes, *PAIR1* and *RPA2A*, was separately plotted. Error bars indicate SD of three biological replicates. Student’s paired *t*-test: * *p* < 0.05, ** *p* < 0.01.

**Figure 10 plants-11-02899-f010:**
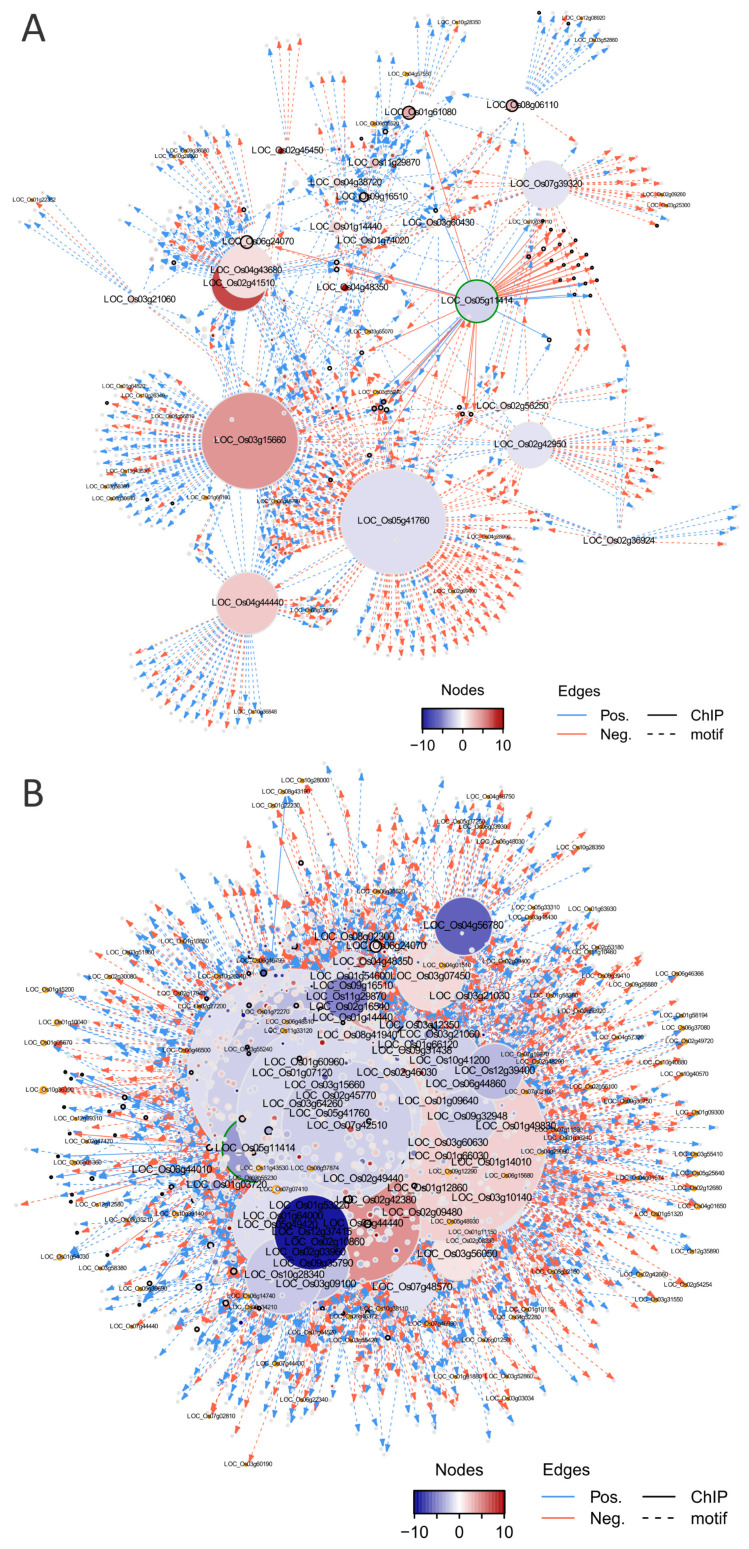
Transcriptional regulatory networks related to multiple pathways. Only the upstream and downstream genes of pathway-related genes differentially expressed in the WT-dSpm (**A**) and WT-CRISPR (**B**) comparison are shown. High quality original files of A and B can be found in the Appendix A.

## Data Availability

All data supporting the findings of this study are available within the paper and within its Appendix A published online. The RNA-seq raw data are available in the NCBI Gene expression Omnibus database (GEO) with accession number GSE181713.

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
