# Peer review of "OsMADS58 Stabilizes Gene Regulatory Circuits during Rice Stamen Development"

_plants, 2022, doi:10.3390/plants11212899_

Round 1
Reviewer 1 Report
The paper entitled “OsMADS58 stabilizes gene regulatory circuits during rice stamen development ” by Liping Shen et al described the functional characterization of OsMADS58 in stamen development using newly developed CRISPR KO line. Although the phenotype of the transposon insertion mutant of OsMADS58 had been previously analyzed, it was not a null mutant and its detailed function remained unclear. The advance in this study is the creation of a null mutant for OsMADS58 by CRISPR. The authors have conducted histological and transcriptome research and revealed the function of the gene in early stamen development.
I think that the paper is sufficient to merit publication in Plants though a major revision is recommended which needs to include the following points.
(1) The description of the results is not concrete and sufficient. Vague and ambiguous expressions such as “abnormal” and “abberant” are often used. The results should be explained as carefully as possible.
(2) Discussion only repeats results and gives no useful information to the reader. It should be completely rewritten.
(3) Some of the figures are small and low resolution.
(4) Line 80: 2.1 Subsection
(5) Line 112: What does “abnormal” mean? Shape, number, size, or other characters?
(6) Line 118: How is meiosis “disturbed”? Describe differences from WT. Why does disturbing of meiosis lead increase of cell number (not decrease)? It is need further explanation.
(7) Line 153: Number of biological replicates for osmads58 CRISPR lines is 2. Generally, the number of biological replicates should be at least 3. I believe that the present analysis provides sufficient insight, but the reason of the small number of replicates and the limited interpretation of the results should be clearly stated in the paper.
(8) Line 153: Although text showed that nine RNA-seq sample was analyzed, only six samples are found in figure9A (and tables). If proper reason and criteria of exclusion of samples are not explained, it is regarded as wrong arbitrary data manipulation.
(9) Line 161: Show total number of target genes and number of upregulated among them in dSpm and CRISPR line.
(10) Line 175: To show “clearly distinguishable” and similarity between WT and dSpm, sample clustering based on expression profile is needed.
(11) Figure 3E: Use box plots of dot plots to clearly show the distribution pattern.
(12) Figure 8: I can not see the results of the network analysis clearly. The figure needs to be presented in a different way.
Reviewer 2 Report
The manuscript “OsMADS58 stabilizes gene regulatory circuits during rice stamen development” by Shen and co-workers presents the results of continuing studies to elucidate the role(s) of OSMADS58, a C-class MADS box protein in rice. The authors generated and characterized a CRISPR knockout mutant (osmads58) that exhibits complex phenotypic alterations in the stamen, including extra diploid germ cells, aberrant meiosis, and delayed tapetum degeneration. The CRISPR mutant line exhibited more drastic changes in both phenotypic stamen alterations and in the expression of OsMADS58 target genes than a previously described osmads58 dSpm transposon insertion line. Detailed RNA-seq analyses on young panicles containing stage 4–6 stamens from the wild type, dSpm, and CRISPR lines identified altered gene expression networks. A transcriptional regulatory network analysis centered on OSMADS58 showed that transcriptional regulatory circuits were substantially altered in the CRISPR line compared to wild type and the dSpm line. Based on the results of their work the authors propose that OsMADS58 has a potential role in stabilizing gene regulatory circuits during early stamen development.
The manuscript is generally well written and on an interesting and important topic in that it attempts to understand the interlinked transcriptional regulatory networks that control stamen development, including germ cell differentiation, meiosis, and pollen development. It presents a considerable amount of data, which is of high quality and generally supports the conclusions that are drawn. However, I do have several concerns/suggestions regarding the manuscript.
1. Line 112: Did the authors analyze independent CRISPR lines as written or individual plants from one CRSPR line? Given differences between the dSpm and CRISPR lines a preliminary phenotypic characterization of multiple independent lines that demonstrated similar phenotypes would be ideal. But at the least the authors need to clarify what specifically they analyzed.
2. The sentence in lines 119 &120 should probably be deleted, this is speculative and belongs in the discussion, if anywhere.
3. The authors should clarify and document whether the first meiotic defect was observed at diakinesis. Did they do a complete meiotic analysis and if so, are they normal before this? These data should be shown in the Supplementary Materials. Likewise, the authors indicate that meiosis in the osmads58 CRISPR resembles that of the rice mutants in the meiotic genes OsSDS, CRC1, PAIR2, OsSPO11-1, OsMTOPVIB, and P31 comet. If I am not mistaken these mutants show somewhat different meiotic phenotypes. It may be that diakinesis is altered in all the mutants, but that is likely not the primary defect. The authors need to be more precise in their language.
4. Figure 8 is impossible to read and/or follow. Other than being able to tell that the images on the left are different than those on the right, it is impossible to discern anything from Figure 8. While Figure 10 is somewhat better, no real meaningful information can be drawn from the image other than the networks are complex and different.
Round 2
Reviewer 1 Report
The authors appropriately addressed all of my comments. I think that the paper is ready to go.
Author Response
Thank you for your appreciation of our efforts on the revision and granted your full support for the publication of our work on the prestige journal of Plants.